# Spatial Modes of Laser-Induced Mass Transfer in Micro-Gaps

**Tobias C. Foertsch [1], Alex T. Davis [1], Roman Popov [1], Clemens von Bojničić-Kninski [1], Felix E. Held [2], Svetlana B. Tsogoeva [2], Felix F. Loeffler [3] and Alexander Nesterov-Mueller [1,*]**

[1]  Institute of Microstructure Technology, Karlsruhe Institute of Technology, Hermann-von-Helmholtz-Platz 1, 76344 Eggenstein-Leopoldshafen, Germany; mail@tobias-foertsch.de (T.C.F.); alex32davis@yahoo.com (A.T.D.); roman.popov@kit.edu (R.P.); clemens.bojnicic-kninski@kit.edu (C.v.B.-K.)

[2]  Institute of Organic Chemistry I, Friedrich-Alexander-Universität Erlangen-Nürnberg, Nikolaus-Fiebiger-Straße 10, 91058 Erlangen, Germany; felix-held@gmx.de (F.E.H.); svetlana.tsogoeva@fau.de (S.B.T.)

[3]  Max Planck Institute of Colloids and Interfaces, Department of Biomolecular Systems, Am Mühlenberg 1, 14476 Potsdam, Germany; Felix.Loeffler@mpikg.mpg.de

*   Correspondence: alexander.nesterov-mueller@kit.edu

**Abstract:** We have observed the concentric deposition patterns of small molecules transferred by means of laser-induced forward transfer (LIFT). The patterns comprised different parts whose presence changed with the experimental constraints in a mode-like fashion. In experiments, we studied this previously unknown phenomenon and derived model assumptions for its emergence. We identified aerosol micro-flow and geometric confinement as the mechanism behind the mass transfer and the cause of the concentric patterns. We validated our model using a simulation.

**Keywords:** LIFT; bio-printing; aerosol; confinement; deposition patterns

## 1. Introduction

We observed concentric deposition patterns (Figure 1) during our study of the laser-induced forward transfer (LIFT) process for biochemical compounds [1]. These patterns comprised an inner part and several outer parts whose presence changed with the process parameters. We studied this phenomenon in experiments varying the geometric conditions and developed a model that reproduced the main characteristics in numerical simulations. In this paper, we present our findings and possible explanations for these observations.

LIFT denotes a class of processes [2], where a laser beam induces mass transfer, mainly indirectly due to heat. The variants of LIFT reported in literature differ in the mechanisms that cause the mass transfer, such as ablation [3], droplet formation from a solid phase [4], droplet formation from a liquid phase [5], mechanical actuation due to the formation of blisters [6], and so forth. For our implementation of LIFT, we used a thick polymer layer (2 in Figure 1a), adhered to a glass substrate for absorbing a deflected, focused laser beam. Material was transferred from a second thin polymer layer, which embeds biomolecules (3 in Figure 1a) across a gap (4 in Figure 1a) onto a substrate that is able to bind the molecules (5 in Figure 1a). The gap between the two substrates lying on top of each other was caused by the waviness of the substrates and was on the order of tens of microns. The relevant physical effects leading to the mass transfer in our implementation of LIFT was previously unclear, and its determination was part of this study. We concluded that an aerosol was created and that it was subject to confinement conditions in the gap. This diverges from the known transfer mechanisms mentioned above. After chemical processing and fluorescent staining, images of the deposition patterns could be taken. Their intensity corresponded to the surface concentration of the bound molecules.

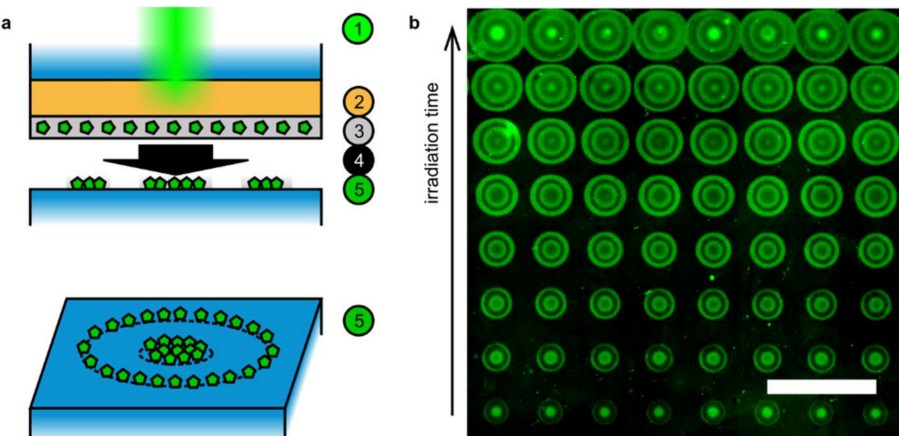

**Figure 1.** Experimental scheme: (**a**) A focused laser beam (1) heated an absorption layer (2) and a coated polymer thin film containing the compounds to be transferred (3). This generated a micro-jet (4) confined between the two plates, probably by evaporating solvent residues and creating an aerosol. Concentric deposition patterns (5) were observed thereafter on the acceptor slide. (**b**) Image of self-assembled concentric amino acid patterns (fluorescently labeled), as transferred onto the acceptor slide by the laser process. Increasing irradiation time (1, 1.5, 2, 3, 5, 7.5, 10, and 20 ms) resulted in the occurrence of additional concentric rings of the deposited material. The scale bar represents 1 mm. The laser beam diameter was 15 μm.

## 2. Experimental Setup Details and General Procedure

Our applications of LIFT comprise the combinatorial synthesis of dense peptide arrays from amino acid building blocks, which was shown elsewhere [1]. This underpins the specific architecture of the coated surfaces and the chemical processing steps described below. The steps followed the Merrifield solid-phase peptide synthesis [7], which are described in [1]. Chemicals were purchased from VWR International, Missouri, TX, USA, Sigma-Aldrich, St. Louis City, MO, USA and Merck KgaA, Darmstadt, Germany.

### 2.1. LIFT Surfaces

The upper surface (donor slide) consisted of a microscope glass slide (Paul Marienfeld GmbH & Co. KG, Lauda-Königshofen, Germany) as the substrate, covered with a light-absorbing self-adhesive polyimide film (70110, CMC Klebetechnik GmbH, Frankenthal, Germany; comprising 25 μm Kapton® HN, DuPont, St Joseph, MO, USA) and spin-coated (SCC-200, Schaefer Technologie GmbH, Langen, Germany) with a styrene-acrylic copolymer (SLEC PLT 7552, Sekisui Chemical GmbH, Germany; 135 mg/slide in 1 mL dichloromethane; rotation speed 80 rps for 40 s, resulting in a thickness of typically 1 μm) embedding fluorenylmethoxycarbonyl-glycine-pentafluorophenyl esters (Fmoc-Gly-OPfp; Sigma-Aldrich, St. LouisCity, MO, USA and Merck Schuchardt OHG, Hohenbrunn, Germany; 15 mg/slide). Glycine is an amino acid and Fmoc is the protection group for the terminal amino group.

The lower surface (acceptor slide) was a glass slide functionalized with a polymer layer (10:90 amino-terminated poly(ethylene glycol)methacrylate and methyl methacrylate) bearing free amino groups (PEPperPRINT GmbH, Heidelberg, Germany). Acceptor slides with dry-etched micro-cavities were also used in Reference [8].

The spherical particles used as obstacles in Figure 2c–e were commercially available cross-linked poly(methyl methacrylate) spheres (Spheromers CA-10, Microbeads, Skedsmokorset, Norway).

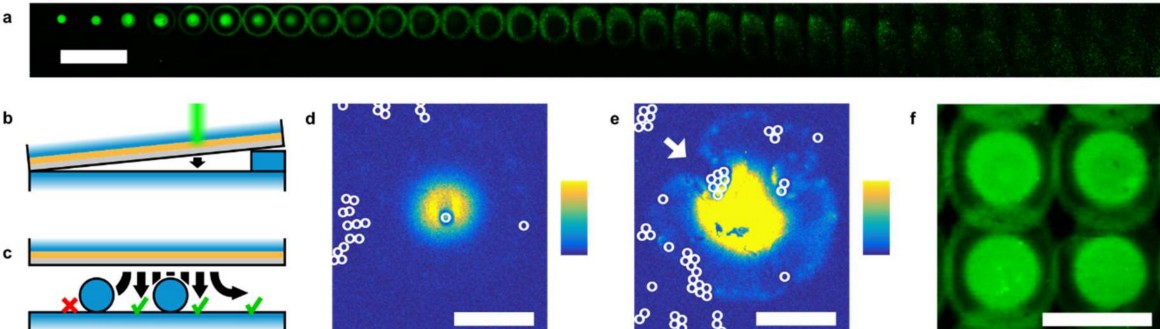

**Figure 2.** (**a**) Fluorescent image of deposition patterns with increasing distance (shown here up to 580 μm) between the donor and acceptor slides (a sketch of the experiment is shown in (**b**)): A ring occurred around the spot while its core was vanishing. For large distances (right end of the fluorescence image), the ring drifted away from the irradiated location. The scale bar represents 1 mm. (**c**) Scheme of an experiment to determine the flow direction in the gap: obstacles, here micro-spheres (diameter 10 μm), could block the lateral flow. (**d**,**e**) Fluorescent images of the deposition pattern blocked by micro-spheres (individual pseudo-colors for increased contrast; scale bar 100 μm). (**f**) Fluorescent image of overlapping patterns. The length of the scale bar is 250 μm.

### 2.2. Laser System

The material transfer from the donor slide to the acceptor slide was actuated by a laser beam from a 1 W 532 nm continuous wave diode-pumped solid-state laser (FSDL-532-1000T, Frankfurt Laser Company, Friedrichsdorf, Germany) that was modulated by an acousto-optic modulator (002AF1, Polytec GmbH, Waldbronn, Germany) into pulses on the order of milliseconds and deflected in a scan head (hurrySCAN 10, Scanlab AG, Puchheim, Germany) with a 100-mm focusing f-θ-lens. The laser radiation was absorbed by the polyimide film, and heated the polymer layer with the embedded amino acids and induced the material transfer across the gap, probably by evaporating solvent residues and creating an aerosol.

### 2.3. Chemical Processing

After the transfer, the activated monomers were coupled to the acceptor slide in a separate heating step (90 °C for 60 min inside a custom-made sealed steel chamber under an argon atmosphere). Uncoupled material was subsequently washed from the acceptor slide in a custom-made automated reactor device. There, the unreacted amines to which no compound had coupled were capped via an acetylation (blocking of $NH_2$ groups of the surface by 10% *v/v* acetic anhydrate and 20% *v/v* N,N-diisopropylethylamine in 70% *v/v* N,N-dimethylformamide with ultrasound for 3 min, and once again for 15 min; washing with acetone for 1 min without and 1 min with ultrasound, and once again with ultrasound for 1 min). In addition, the Fmoc groups were removed (deprotection of the $NH_2$ groups of the coupled amino acids by 20% *v/v* piperidine in 80% *v/v* N,N-dimethylformamide for 1 min with and for 15 min without ultrasound; washing with acetone for 1 min without and for 1 min with ultrasound, and once again for 1 min with ultrasound).

### 2.4. Staining and Imaging

The slide was washed in phosphate-buffered saline with 0.05% Tween 20 (PBS-T) for 10 min, and then the deprotected free amino acid groups of the transferred compounds were fluorescently stained by incubation with 0.05 μg/mL of a rhodamine N-hydroxysuccinimide ester (5/6-carboxy-tetramethyl-rhodamine succinimidyl ester) in PBS-T for 45 min; afterwards, the slide was washed three times for 3 min with PBS-T and purged with pure water.

Fluorescent images were taken at the wavelength of 532 nm with a fluorescence scanner (InnoScan 1100 AL, Innopsys, Carbonne, France) to assess the deposition patterns. For the experiment shown in Figure 3, a previously reported fluorescent compound [9] instead of an amino acid was

transferred and visualized directly; the picture was illuminated by a UV lamp (UVGL-58, UVP, Jena, Germany) and taken by a camera system equipped with suitable filters (see [9] for details). Evaluation and linear contrast adaptions of the pictures were done in ImageJ and MATLAB (The MathWorks Inc., Natick, MA, USA). Topographical measurements were performed with a vertical scanning interferometer (Contour GT, Bruker, Billerica, MA, USA).

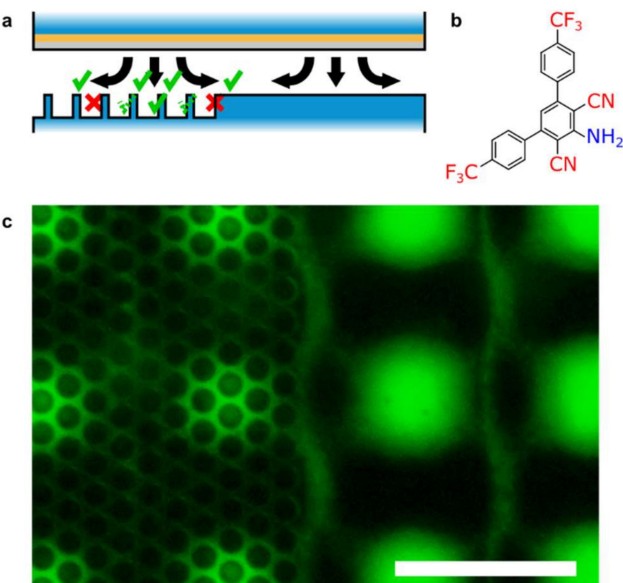

**Figure 3.** Material flow over and into micro-structures. (**a**) Schematic sketch of the experiment; (**b**) Structure of the applied fluorescent compound [9]; (**c**) Actual fluorescent image of the deposition pattern, in green color scale. The pattern on top of the structured substrate is similar to the result on a flat surface, but the bottom of the micro-wells was reached by the material flow and stained only in the center region of the spots. The diameter of the cavities was 23 µm, their smallest distance 9.5 µm, and their depth 8.7 µm. The scale bar represents 250 µm.

## 3. Details of the Numerical Simulation

Numerical simulations of the material transport were undertaken in COMSOL (COMSOL Inc., Burlington, MA, USA). We assumed the compounds were transported as diluted species in a laminar flow, subject to diffusion. The adsorption to the substrates was modeled with an ordinary differential equation defined on the boundaries and by using respective boundary conditions for the partial differential equations within the simulation domain.

### 3.1. Geometry

Due to the axis-symmetric nature of the problem, the domain can be a rectangle bounded on top and bottom by the substrates, with a sufficiently large radial extent (here 500 µm). The inlet of the flow advecting the compounds was at the top in the lateral center of the domain. At the outer end, the domain was open for the flow. We varied the inlet conditions and the height $H$ of the domain and studied the effect on the radial distribution of the adsorbed surface concentration after the transfer, which is the measurand that was comparable to the experimental observations.

### 3.2. System of Equations

The governing equations of our model were the Navier–Stokes equations for Newtonian fluids $\rho \partial u / \partial t + \rho (u \cdot \nabla) u = \nabla \cdot \left[ -pI + \mu \left[ \nabla u + [\nabla u]^{\mathrm{T}} \right] \right]$ with the incompressibility condition $\rho \nabla \cdot (u) = 0$, where $u(r, z, t)$ is the velocity field (initially 0 everywhere), $p(r, z, t)$ the pressure, $\rho = 1.636 \mathrm{kg/m}^3$ the density, and $\mu = 18 \cdot 10^{-6}$ Pa·s the viscosity of air, and the advection–diffusion equation $\partial c / \partial t + \nabla \cdot (-D \nabla c) + u \cdot \nabla c = 0$ for the volume concentration $c(r, z, t)$ (initially 0), with the diffusion coefficient

$D = 10^{-7}$ m$^2$/s that was chosen to be as large as is reasonable for molecules in a gas. From these material parameters, the characteristic pattern size of 100 μm and the inlet velocity taken as 1.5 m/s resulted in a Reynolds number $Re \approx 14$, whereas the critical Reynolds number for 2D flows between parallel plates was larger than 2285 [10]. If the inlet diameter $2R = 2 \cdot 37.5$ μm was used as length scale for the calculation, 10 was obtained as the value of $Re$; there was one steady-state study of radial entrance flow between plates [11] that showed laminarity for values much higher than that (the series ended at $Re = 200$). These considerations justify why we have not introduced a turbulence model term. The surface concentration $c_s(r)$ (initially 0) was integrated by the equation $\partial c_s / \partial t = |kc|$ defined on the boundary, where $k = 20,000$ m/s was a very high mass transfer coefficient ensuring complete deposition of the compounds. Applying the absolute value favored numerical stability.

### 3.3. Boundary Conditions

Accordingly, an outward flux $-\boldsymbol{n} \cdot (-D\nabla c + \boldsymbol{u}c) = -kc$ was set as the boundary condition for the domain equations ($\boldsymbol{n}$ denotes the normal vector on the boundary). Inflow of the fluid and of the diluted species took place at $0 < r < R$, $R = 37.5$ μm, for $0 < t < \tau$, $\tau = 1$ ms, and $0 < t < \varepsilon\tau$, $\varepsilon = 1/2$, respectively. The fluid entered normal to the boundary with $|\boldsymbol{u}| = U$. The inlet concentration $C$ of about 0.45 mol/m$^3$ was calculated such that for $U = 1$ m/s, one obtained 1 pmol of compounds in total. For numerical stability, these values were not set instantaneously, but were approached following a logistic function with a small time scale $\chi = 0.01$ ms $\ll \tau$: $\sim 1/(1 + \exp((t - t_{\text{start}})/\chi)) \cdot (1 - 1/(1 + \exp((t - t_{\text{end}})/\chi)))$. The start time was $5 \cdot \chi$ so that the expression was nearly zero for $t = 0$. The other parts of the top and bottom boundaries imposed the no-slip condition $\boldsymbol{u} = 0$ for the velocity field and the flux condition for the volume concentration due to adsorption mentioned above. The outlet conditions at the right boundary were a fixed zero pressure with suppressed backflow, $\left[-pI + \mu\left[\nabla\boldsymbol{u} + [\nabla\boldsymbol{u}]^{\text{T}}\right]\right]\boldsymbol{n} = -\hat{p}_0\boldsymbol{n}$ with $\hat{p}_0 \leq 0$ where $I$ was the identity matrix, and $-\boldsymbol{n} \cdot D\nabla c = 0$.

### 3.4. Solver Details

The domain was discretized into a free triangular mesh by COMSOL with "finer" element size and "physics-controlled" sequence type, resulting in elements with an average area of 54.8 μm$^2$. The choice of the mesh size had no tremendous impact on the outcome of the simulation. The solver parameters were left in their default settings ("BDF" time stepping, "constant (Newton)" fully coupled solver with Jacobian update "once per time step", "PARDISO" direct solver). The simulation was run until $t = 50 \cdot \tau$.

For Figure 3c, a two-dimensional image that a fluorescence scan would reveal in experiments was reconstructed in MATLAB from the simulation result $c_s(r)$ of the last time step.

## 4. Results

We observed concentric patterns in the fluorescent image of spots of transferred material (Figure 1b), which differed depending on the LIFT irradiation time.

To explain the mechanism behind the generation of the patterns, we investigated the dependence of the deposition pattern on the geometric confinement, as shown in Figure 2a,b. The increasing distance between the two planes, achieved by a spacer inserted between them at one end, induced the appearance of different deposition "modes": the center disappeared while the circle became more pronounced. Beyond transfer distances of approximately 300 μm, the spot locations drifted from the axis of the laser beam and the spots transformed into a blurred trace. This led us to assume that the mass flow was splitting into two motions: one was perpendicular to the donor surface causing the center of the pattern; the other motion propagated along the acceptor surface, causing the ring, and was able to drift. The drifting and blurring indicate an aerosol-like behavior of the transferred matter.

The material movement parallel to the slide surface could be demonstrated more clearly if the material was transferred to a surface contaminated with spherical particles (Figure 2c–e). Particles

scattered in the margin regions of the pattern prohibited the material transfer in the lateral direction, so that deposition "shadows" appeared in the radial direction.

Figure 2f illustrates the additive character of the mass transfer: overlapping parts of the pattern possessed fluorescent intensities that were a factor of 2.2 higher than those in non-overlapping areas. It can be concluded that the dark region between the central core and the outer ring of a spot was not merely a destruction artifact of the acceptor surface, because the compounds within the ring region of the neighboring spots could still successfully couple to these areas.

To prove the existence of lateral and vertical deposition modes, we also studied the deposition of material patterns onto hexagonally arrayed cylindrical micro-cavities (Figure 3). The bottoms of the micro-structures showed strong signals only in the center of the deposition pattern $I_b^c / I_t^c = 0.71$, whereas the cavities of the outer regions were stained significantly weaker $I_b^p / I_t^p = 0.47$. Here $I_b^c / I_t^c$ is the ratio of the fluorescent signal mean intensities at the bottom of the cavities and at the top surface in the center of the deposition pattern, and $I_b^p / I_t^p$ is the same ratio in the outer regions. This implies that the central point was exposed to a significantly stronger vertical material flow. The lateral material transfer, which could not enter the micro-structures, dominated in the margin area.

To test our theory and to understand the phenomenon, we simulated the mass transfer induced by laser radiation under confinement using an advection-diffusion model coupled with a boundary equation integrating the deposited surface concentration. From our experimental findings described above, we derived the model assumption that the compounds were transported as diluted species in a laminar flow and were subject to diffusion. The inlet of the flow transporting the compounds by advection was at the top boundary of the gap, where the laser deposited heat into the donor slide, evaporated and expanded solvent residues and thereby created an aerosol (Figure 4a). A main assumption in the model was that the concentration of compounds in the inflow was maximal at the beginning of the process and low after a certain time, when all accessible compounds were entrained (Figure 4b). By integrating the amount of material adsorbed to the lower boundary of the gap, we reconstructed the observable patterns of the molecules deposited on the acceptor slide (Figure 4c–e).

In the scope of this model, we simulated deposition patterns (Figure 4c) similar to those observed in experiments. The pattern on the acceptor slide surface formed sequentially; first the core was formed as a result of the perpendicular jet impingement, then the concentric ring formed when the compounds diffused as a ring-shaped cloud slowly across the substrates.

This model could successfully reproduce the experimental findings from Figure 2a,b, where an increase of the distance between donor and acceptor slides led to a smooth conversion of modes from a vanishing central core to a growing ring (Figure 4d). In addition, the variation of the inlet velocity in the simulation influenced the deposition patterns in a similar way: increasing the inlet velocity eventually led to a deposition minimum in the central point (Figure 4e). However, it should be mentioned that the simulations were consistent with the observations only within a subset of the entire parameter space (the dashed line in Figure 4d, corresponding to the narrowest gap, reveals the largest ring, contrary to the trend). Nevertheless, the simulations predicted and confirmed the two-fold perpendicular and lateral deposition mechanism.

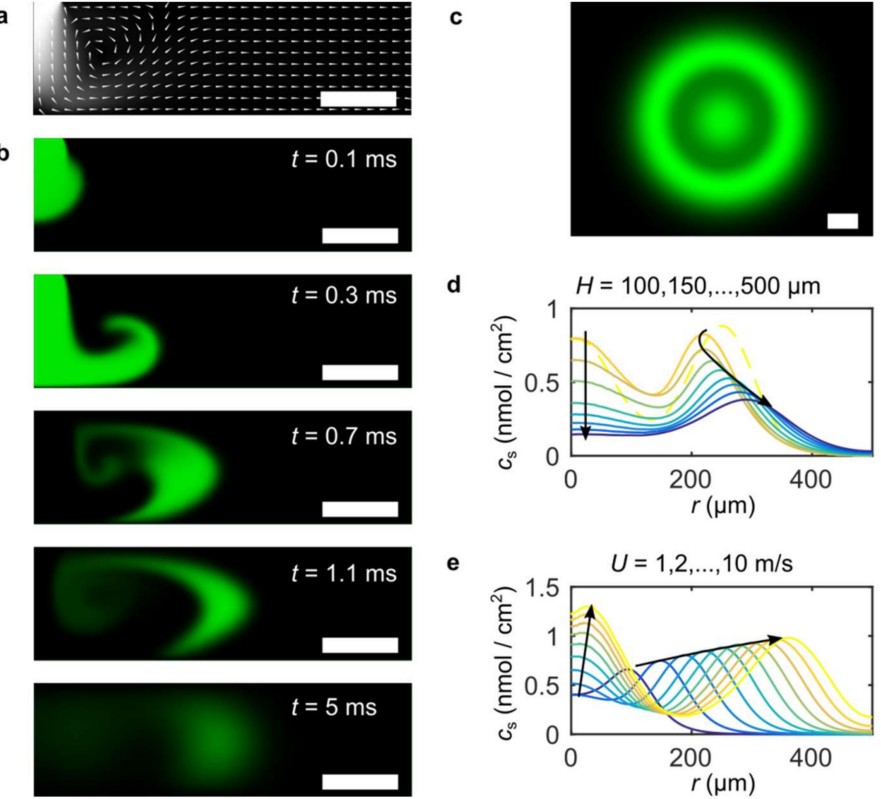

**Figure 4.** Results of numerical modelling. (**a**) Side view of the quasi-stationary flow field (gray scale: absolute value, arrows: direction) generated in the micro-gap by the evaporation of solvent residues from the material layer during laser irradiation. The left side of the picture corresponds to $r = 0$, the donor substrate is located on the top of the domain, the acceptor on the bottom; the right end is open. (**b**) Snapshots of the volume concentration (side view) during different flow phases: initially, the inflow jet (5 m/s) carrying the compounds hit the acceptor surface and turned radially outward (first and second picture). In a second phase (here, between $t = 0.5$ ms and $t = 1$ ms ), the inflow continued but did not bring in any more new compounds (third picture), until the flow stopped completely. Then, the compounds formed a ring-shaped cloud that diffused slowly to the substrates (fourth and fifth pictures). The scale bars represent 100 μm. (**c**) Reconstructed deposition pattern (top view) resulting from the simulation shown in (**b**). The spot exhibits a center core and a ring. The scale bar represents 100 μm. (**d**) Radial surface concentration profiles of the simulated deposition patterns for different distances between the substrates. As indicated by the eye guiding arrows, the core of the spot was vanishing, and the ring was decreasing more slowly and moving slightly outwards. The height of the gap simulated in (**b**) was 150 μm; (**e**) Simulated radial profiles of the spots for different inlet velocities; the arrows illustrate that the core deposition was stronger with respect to the ring formation when the jet transfer was faster. Furthermore, for higher velocities, there was a growing dimple at the center. As expected, faster flows carried the ring farther away from the center. Profiles in (**d**) and (**e**) correspond to the time $t = 5$ ms.

## 5. Discussion and Conclusions

We observed concentric deposition patterns of small molecules transferred by means of LIFT. From our experimental study, we learned that the presence of the different parts of these patterns was more or less pronounced in a mode-like way, depending on the experimental parameters. No significant deformations of the donor slide after irradiation were observed, so we excluded the mechanical transfer of the molecules to the donor slide by its direct contact with the acceptor slide. From the experimental findings, the model assumption of an aerosol-driven mass transfer was derived and tested in a numerical simulation. The simulation could reproduce the observed phenomena, and thus validated

the model. Its time-resolved results provide insight into the confined flow within the micro-gap and enable a deeper understanding of the observed transfer modes.

It is unlikely that optical effects such as different Laguerre-Gaussian laser modes within the coating layers were the cause of the concentricity of the deposition patterns. This is because the diameter of the laser beam was significantly smaller than the diameter of the concentric rings and their size increased gradually with irradiation time. The time-transient behavior is especially difficult to explain. Another optical explanation could be the diffraction of the laser beam into the (off-focus) intensity distribution of the Airy point spread function. However, it is unclear how this would be caused and which mechanism would reflect this in a corresponding deposition pattern.

Although laser-induced material transfer has been known for several decades, to our best knowledge the existence of such transfer or deposition modes has not yet been observed. We argue that this is because in our application and work with biomolecules, we use minute amounts of material and are able to detect them, respectively. Conducting literature research, we found a related effect in an early LIFT publication [12] studying transfer by a laser with linear focus. There, instead of one line, two parallel lines, and when the intensity was increased, three lines of deposited material were reported by the authors. They found a thermal explanation and considered the phenomenon as a disturbing effect on the material deposition, without studying it in greater detail. Indeed, in many applications one tries to avoid possible contaminations or confined geometries. Furthermore, in many applications, low contamination amounts do not become visible. Therefore, the deposition modes were hidden to the latter experimentalists.

**Author Contributions:** Conceptualization, investigation, methodology, visualization, formal analysis, writing—original draft, T.C.F.; software, investigation, A.T.D.; resources, investigation, R.P.; resources, investigation, C.v.B.-K.; resources, investigation, F.E.H.; resources, supervision, funding acquisition, S.B.T.; supervision, writing—review and editing, F.F.L.; supervision, methodology, writing—original draft, funding acquisition, A.N.-M.

**Funding:** This research was funded by the ERC, St Grant no. 277863, PoC no. 766695 and by the Deutsche Forschungsgemeinschaft (DFG), TS87/15-1, TS87/17-1.

**Acknowledgments:** We would like to thank D. S. Mattes, J. Greifenstein, and D. Haeringer for their technical support.

**Conflicts of Interest:** A.N.-M., F.F.L., and C.v.B.-K. are named on pending patent applications relating to molecule array synthesis (application number PCT/EP2013/001141, PCT/EP2014/001046, and US Patent Application 20160082406). All other authors declare no conflict of interest. The funders had no role in the design of the study; in the collection, analyses, or interpretation of data; in the writing of the manuscript, or in the decision to publish the results.

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
