# Peer review of "Spatial Modes of Laser-Induced Mass Transfer in Micro-Gaps"

_applsci, doi:10.3390/app9071303_

Round 1

Reviewer 1 Report

My main question, the study of a blister deformation that could affect the transference dynamics has been answered. "No significant deformations of the donor slide after irradiation were observed so that we exclude the mechanical transfer of the molecules to the donor slide by its direct contact with the acceptor slide."

This deformation could take place in the us range and the later recovery of the donor could make the deformation non significant. Experiments with high speed imaging would be a good way to address this possibility.

I think that the paper should be published in this form and let this question be solved later with different works.

Reviewer 2 Report

The questions from the previous round have been answered clearly.

This manuscript is a resubmission of an earlier submission. The following is a list of the peer review reports and author responses from that submission.

Round 1

Reviewer 1 Report

This paper is very interesting and it provides very useful information to those working on LIFT methods for printing of substances in general. It is also useful for those working on printing biomolecules and sensitive materials. It is well written, provides a well-described simulation of the process, describing and giving interesting results regarding the deposition parameters and phenomena observed. It discusses the results concisely, but, in my opinion, with a solid argument. For these reasons I recommend it for publication in its present form. The authors may take into account the comments included in the pdf enclosed.

Reviewer 2 Report

"Spatial modes of laser-induced mass transfer in microgaps" reports their finding on the concentric deposition patterns of small molecules transferred by LIFT process, and analyzed it from the perspective of aerosol microflow and geometric confinement. This is a very interesting work, however, the reviewer thinks several points in the manuscript is not clear. Here are some of the comments for the revision:

Introduction part states that the concentric deposition is found during their study, yet it might be more helpful if the authors can kindly summarize some of the previous works on the LIFT process.

The authors state that the Reynolds number of the system under concern is Re~14, and could you explain how the value is calculated?

The explanation for Fig. 3 seems to be unclear: the fluorescent signal from the cylindrical microcavities seem to be lowered even when the cavity is directly below the center of deposition, which makes the authors' comment less plausible. It might help if the authors can provide more quantitative data. The same point also applies to Fig. 2f: the manuscript states that the pattern gives more intensive fluorescence signal, which seems to be not clear from the picture only.

4. The results are not arranged consistently: Fig. 1 shows the fluorescent image when the irradiation time is altered; Fig. 2 for different gap size, while Fig. 4 analyzed the data for the gap size change and jet (inlet) velocities. The authors should be clear on the relation between irradiation time and the inlet velocity.

The reviewer is not sure if these patterns are changing in a mode-like way since the deposition pattern changes continuously according to the laser parameters, and there are no clear distinct modes. (e.g. different laser modes such as TEM00 and so on.)

Reviewer 3 Report

The authors have studied a mass transfer process during laser-induced forward transfer.  The experimental study revealed an interesting mass-transfer phenomenon in which small fluorescent molecules are deposited in manner that depends upon the separation between the donor and acceptor substrates.  The experiments are clearly described and characterized.  The introduction is rather brief and doesn’t cover much of the literature on laser-induced forward transfer, which is quite extensive now. My primary concerns are the details of the simulations and their interpretations, which are as follows:

1.     The density of the air used in the simulation (1.636 kg/m3) is rather high, and this value seems consistent with air at low temperatures, somewhere in the 200-250 K range.  The viscosity however, seems to be the correct value at about 300 K.

2.     In describing the numerical model, the authors have calculated a Reynolds number, Re = 14, and state the flow is laminar since it is less than the critical value for flow between parallel plates.  The length scale and velocity used for this calculation are not clear.  The experimental configuration is quite different from parallel plate flow, since the flow is actually radial and flows outward from the centerline.  I believe their flow configuration is much more similar to a pulsed jet flow.

3.     The boundary condition for the surface of the acceptor substrate does not seem reasonable, since there is no kinetic or thermodynamic relation between the concentration in the gas phase and the adsorbed concentration of the molecules. The assumed mass transfer coefficient of 20,000 m/s also seems extremely high.  I am not convinced that their mass transfer deposition calculations are reasonable.  They may capture some of the experimental behavior, but not in a realistic way.

4.     In Figure 4d the results do not appear to agree with experimental results shown in Figure 2a very well.  Figure 2a shows that the concentration of the deposited molecules at r = 0 decreases to very small values (nearly zero) as the gap thickness increases.  Figure 4d shows that the predicted concentration decreases slowly to a fixed non-zero value.  Even though the model correctly captures some of the patterns from the experiments, the physical interpretation of the deposition process and the connection to the flow patterns in Figure 4a,b is not obvious to me.  Perhaps the molecules cool down and do not stick as well as the gap increases?  Maybe the ring pattern is influenced by the vortex roll-up?  Maybe the dark ring is due to molecules re-evaporating due to some sort of flow behavior.

5.     Figure 4d,e should have legends to indicate the times of each curve, for better comparison to experiments.

Reviewer 4 Report

I find this study really interesting. The authors present a deeper study of their CW laser LIFT technique trying to understand the physical processes that take place during the transference process.

I would like the author to include information about the interaction of the laser with the thick polymer layer (that includes an adhesive). Is this layer just acting as a heat transfer layer? Is there any final deformation due to the generation of  a vapour pocket in the glass/polymer interface? In your previous paper (ref 1) some images of this deformation (with ring shapes)  were shown at high energy, is this a vibration mode of the polymer?, is there a transition between low energy elastic deformation and high energy plastic deformation of the polymer?, could the modes be related to vibration modes?

If this has been studied and you have seen that this does not have any influence in the process, some discussion should be added to the paper.

Some other minor suggestions are included for each section.

-Introduction

Some references to LIFT processes that share some characteristics (like Blister actuated LIFT or absorbing layer LIFT) could be added and their differences explained.

Details on the numerical Simulation - Boundary conditions

I think this is the key parameter in the model, how the interaction is included to start the fluid motion. In this case the fluid motion at the polymer/fluid interface is imposed as a velocity field. Some graphs to visualise the values used in the simulations and some discussion about this model is needed.

-Results

In figure 2a it is explained that the ring drifts away from the irradiated location, but I can not understand the explanation given to this phenomenon. A possible explanation could be some misalignment effect that produces a flow that it is not normal to the acceptor surface and this is more clearly shown as the gap increases.

Some experiments with an increased distance between the spot would help to understand the physics in the individual transference process.

-Discussion and Conclusions

I alike the discussion about optical effects and I agree with the authors in their conclusions.

Here I would like to see the discussion about the polymer deformation (electric/plastic deformation) and its possible impact in the study.

I think that with this information the paper would be improved and most possible explanations would be studied.

This research is worth publishing once this questions are addressed.